# Effectiveness of a Combined Toothbrushing Technique on Cariogenic Dental Biofilm in Relation to Stainless Steel and Elastomeric Ligatures in Orthodontic Patients: A Randomized Clinical Trial

**DOI:** 10.3390/healthcare11050731

**Published:** 2023-03-02

**Authors:** Thanakorn Saengphen, Sittichai Koontongkaew, Kusumawadee Utispan

**Affiliations:** 1International College of Dentistry, Walailak University, Bangkok 10300, Thailand; 2Faculty of Dentistry, Thammasat University, Pathum Thani 12120, Thailand

**Keywords:** combined horizontal–Charters–modified Bass tooth brushing, dental biofilm, dental caries, elastomeric ligature (EL), randomized clinical trial, stainless steel ligature (SSL), three-tone plaque disclosing gel, oral health

## Abstract

Increased dental biofilm commonly occurs during orthodontic treatment. The aim of this study was to evaluate the effect of a combined toothbrushing method on dental biofilm cariogenicity in patients with stainless steel (SSL) and elastomeric (EL) ligatures. At baseline (T1), 70 participants were randomized (1:1 ratio) to the SSL or EL group. Dental biofilm maturity was evaluated using a three-color-disclosing dye. The participants were instructed to brush their teeth using a combined horizontal–Charters–modified Bass technique. Dental biofilm maturity was reassessed at the 4-week follow-up (T2). We found that at T1, new dental biofilm was the highest, followed by mature and cariogenic dental biofilm in the SSL group (*p* < 0.05). In the EL group, cariogenic dental biofilm was highly observed, followed by mature and new dental biofilm (*p* < 0.05). After intervention, cariogenic dental biofilm significantly decreased in both groups (*p* < 0.05). Moreover, a marked decrease in cariogenic dental biofilm was observed in the EL group compared with the SSL group (*p* < 0.05). However, the change in mature dental biofilm in the groups was similar (*p* > 0.05). Our results demonstrated that the combined toothbrushing method reduced cariogenic dental biofilm in the SSL and EL groups.

## 1. Introduction

Biofilms are defined as matrix-embedded microbial populations adherent to each other and/or to surfaces. Dental biofilm or dental plaque is a specific type of biofilm in which a diverse community of oral bacteria is found on the tooth surface as a biofilm [1]. In orthodontic treatment, placing fixed appliances consisting of metals and polymers creates surfaces that differ from those of the natural oral hard and soft surfaces. This creates retention areas in which it is difficult to perform mechanical plaque control [2]. In addition, fixed appliance features increase the amount of dental biofilm and cariogenic bacteria (*Streptococcus mutans* and Lactobacilli) [3,4].

The accumulation of dental biofilm around the brackets is also influenced by the archwire ligation material [5]. Stainless steel ligature (SSL) [6] and elastomeric ligature (EL) [7] are the most commonly used materials to secure the archwire to the bracket slots. SSL demonstrates less dental biofilm retention compared with EL [8,9]. Higher levels of acidogenic bacteria are detected with EL, most noticeably *S. mutans* and Lactobacilli [9,10]. In contrast, other studies found that the two orthodontic ligatures did not cause alterations in the number of *S. mutans* [11,12] and Lactobacilli [12] in dental biofilm around orthodontic brackets.

Patients with fixed orthodontic appliances have difficulty achieving good oral hygiene because the appliances can be a barrier to mechanical cleaning. Food is commonly trapped around the brackets and under the archwires after eating. Moreover, the patients’ lengthy treatment increases the risk of dental caries formation. Therefore, orthodontic patients require enhanced personal oral hygiene programs and regular professional prophylaxis [3,13]. The most effective and frequently used method for dental biofilm control is oral hygiene, consisting of toothbrushing complemented by using dental floss and other supporting dental care. However, in patients with fixed appliances, the procedures become more complex due to the difficulty caused by the appliances. Therefore, conventional toothbrushing techniques need to be modified to determine the best method for orthodontic patients.

Currently, there is insufficient evidence concerning the most effective toothbrushing method for orthodontic patients. Randomized clinical trials that follow the Consolidated Standards of Reporting Trials (CONSORT) guidelines are required to provide adequate-quality evidence of the relative effectiveness of manual toothbrushing techniques [14]. The modified Bass technique is the most commonly recommended technique for controlling dental biofilm [15]. However, few studies have evaluated the effectiveness of toothbrushing on dental biofilm in patients with fixed orthodontic appliances. The horizontal technique, in combination with the Charters brushing techniques, is recommended to clean the bracket sites, whereas the Bass technique is used to remove dental biofilm at gingival sites [16]. Moreover, the scrubbing technique was found to be more effective than the modified Stillman technique, followed by the Bass technique in reducing dental biofilm in orthodontic patients [17].

Little is known about the effectiveness of manual toothbrushing on dental biofilm’s cariogenicity in orthodontic patients. A previous study demonstrated that the modified Bass brushing technique significantly reduced the prevalence of *S. mutans* in the dental biofilm of patients undergoing orthodontic fixed appliances after a 4–12 month follow-up, compared with that of habitual toothbrushing [18]. Powered-toothbrushing combined with antibacterial regimens yielded lower biofilm viability than manual brushing in patients with stainless steel retention wires [19]. 

Most orthodontic trials used the original Silness and Loe plaque index, which does not demonstrate the dental biofilm’s cariogenicity [20]. A three-color disclosing dye (GC TriPlaque ID Gel^TM^, GC Corporation, Tokyo, Japan) has been developed to determine dental biofilm maturity. The three-tone staining color relies on the pH selective response of different dyes, which are rose Bengal (red pigments) and brilliant blue (blue pigments), included in sucrose-containing disclosing liquid, to intraorally reveal the dental biofilm’s age and acid production. The disclosing dye shows newly formed dental biofilm aged less than 1 day old as pink due to the red dye staining on the surface. Older or mature dental biofilms are seen as blue/purple because the blue dye is trapped within the thick dental biofilms that reflect inefficient toothbrushing. Light blue staining indicates areas of cariogenic dental biofilm due to the fermentation of sucrose in the dye resulting in acid production (pH < 4.5) [21]. We previously reported that when using a three-color disclosing gel, high cariogenic dental biofilm was commonly observed in orthodontic patients [22].

Currently, there is no report on the effect of the toothbrushing method on cariogenic dental biofilm removal in orthodontic patients. Therefore, the aim of this study was to evaluate the effect of a combined toothbrushing technique on reducing the cariogenic dental biofilm in orthodontic patients treated with SSL and EL ligatures.

## 2. Materials and Methods

### 2.1. Study Design 

This study was a single-center, open-label, two-arm parallel-group, randomized clinical trial with an allocation rate of 1:1 and was conducted in a private orthodontic clinic in Samut Prakan province, Thailand. The experimental procedures were in accordance with the applicable ethical standards on human experimentation and with the Helsinki Declaration of 1975, as revised in 2013 [23]. The Institutional Review Board at Walailak University approved the study on 13 August 2020 (protocol number: WUEC 20-227-01/2) prior to the experiment. The trial was registered at the Thai clinical trials registry (TCTR20220221003). The data presented in this report adheres to the CONSORT guidelines for reporting trials [24]. All methods were performed per the approved guidelines and regulations.

### 2.2. Participants and Eligibility Criteria

The participants were included from October 25, 2020, to April 7, 2021. The eligibility criteria for the patients comprised: (1) being physically healthy with no medical problems, (2) >12 years old at the commencement of treatment, (3) able to maintain adequate oral hygiene, and (4) being available to follow-up during the 3 months of the study. The exclusion criteria were (1) hyposalivation or (2) antibiotic therapy within 3 months before the trial.

### 2.3. Interventions 

Dental biofilm maturity was assessed at baseline (T1). The patients were treated with straight wire appliances using conventional pre-adjusted brackets (Roth type, Tomy International, Tokyo, Japan). In the SSL group, the patients were ligated with 0.01-inch stainless steel wire (Highland Metals™, Franklin, IN, USA). For the EL group, the patients were ligated with elastomeric rings (Dyna stick, elastomeric ties, DynaFlex company, St. Louis, MO, USA). Both groups were instructed to use a combined horizontal–Charters–modified Bass technique to clean their teeth three times/day (morning, after lunch, and before bedtime) with a manual toothbrush (124 GUM orthodontic toothbrush, Sunstar Corporation, Schaumburg, IL, USA) [16]. The areas around the brackets and archwires on the facial surface were cleaned using the horizontal technique. Next, the Charters technique was used to clean the undercut below and above the wing of the brackets, occlusal, and palatal surfaces, followed by a modified Bass technique to clean the intrasulcular areas. Dental maturity was reassessed after a 4-week follow-up (T2). Blinding of the clinical investigator and participants to the intervention in each group was not possible in this trial.

### 2.4. Outcomes

The primary outcome of this study was dental biofilm maturity. Dental biofilm maturity was measured at baseline (T1) and then re-examined after a 4-week follow-up (T2). The purpose of the study was to compare the change in dental biofilm maturity values after the introduction of a combined toothbrushing protocol in two randomly assigned groups (SSL or EL). 

### 2.5. Sample Size Calculation

The minimum number of participants required was determined using the G*Power version 3.1.9.4 package (Heinrich-Heine University, Dusseldorf, Dusseldorf, Germany). The sample size was estimated based on a medium standardized effect size (Cohen’ d = 0.5) expectation for the changes in the percentage plaque maturity staining (% PMS) between the baseline (T1) and follow-up (T2) in each group [25]. We used the Wilcoxon signed-rank test for T1 and T2 in each group with a power of 0.80 and an error α = 0.05, which determined that a minimum of 28 participants in each group was needed. Considering a dropout rate of 20%, 70 participants were required to achieve a sample size of 35 per group. 

### 2.6. Randomization (Random Number Generation, Allocation Concealment, Implementation)

Randomization of the two intervention groups was performed by a person otherwise not involved in the study. The patients were randomly selected from the eligible patients with a 1:1 allocation rate. The eligible patients were divided into two groups (Group 1 (SSL) and Group 2 (EL)). Simple randomization was performed using the chit-box method to ensure an equal number in each group [26]. We prepared 35 chits by writing No. 1 (for Group 1) on 35 chits and No. 2 (for Group 2) on 35 chits. After folding the chits and putting them in a box and mixing well, the first patient drew a chit and noted the number written on it. The chit was discarded. The next patient drew the second chit first, noted it, discarded it, and the randomization proceeded similarly until the last chit was drawn.

### 2.7. Blinding

Blinding of the clinical investigators and participants to the intervention in each group was not possible in this trial.

### 2.8. Sample Description

Self-reported questionnaires were used to obtain the patient’s sociodemographic data, medical history, dental history, and oral care practices at baseline. Caries risk status was assessed based on sugar and acidic food intake between meals and white spot lesions (WSLs) [27]. Dietary analysis was performed as previously described [28]. WSLs were identified according to Gorelick et al. [29].

### 2.9. Dental Biofilm Maturity

Dental biofilm maturity was assessed by staining all tooth surfaces, except the occlusal surfaces (Figure 1), with GC Tri Plaque ID Gel™, as previously described [30]. Each color-stained dental biofilm type was scored separately, and the most virulent score in each tooth was recorded. Based on the color changes on the tooth surfaces, the % PMS was obtained using the formula:% PMS = (number of teeth with each colored plaque/total number of teeth examined) × 100.

Before the study, an examiner underwent theoretical and practical training in dental biofilm staining [31]. The reproducibility of the measurements was tested by the intraclass correlation coefficient (ICC) [32].

### 2.10. Statistical Analysis

Data analysis was performed using the Statistical Package of Social Sciences version 25 (IBM Corp., Armonk, NY, USA) and GraphPad Prism 7.0 (GraphPad Software, Inc., San Diego, CA, USA). ICC values were calculated to ensure an acceptable level of intra-examiner agreement. First, the normal distribution of the quantitative variables was tested using the Shapiro–Wilk test [33]. The distribution test revealed a non-Gaussian distribution (*p* < 0.05); therefore, we continued using non-parametric tests. Continuous variables are presented as median and quartiles (Q1–Q3), and categorical variables as frequency and percentages [33]. The Mann–Whitney U test was used to compare the baseline values (T1) of % PMS between the SSL and EL groups. The difference in % PMS of the three types of dental biofilm maturity at T1 and T2 in SSL and EL groups was determined using the Kruskal–Wallis with the Dunn–Bonferroni post hoc test. The Wilcoxon signed-rank test was used to compare the pre- and post-treatment % PMS values in each dental biofilm maturity within each group. To compare the post-treatment values between the type of ligation, the Mann–Whitney U test was used [34]. Statistical significance was defined as *p* ≤ 0.05. 

## 3. Results

### 3.1. Baseline Characteristics

Seventy patients (13–53 years old, median 28 years old) were randomly divided into two groups: the SSL (median age 28 years old, range, 13–43 years old) and EL (median age 28 years old, range, 15–53 years old) groups (Figure 2). No patients dropped out due to the adverse effects of the intervention. 

There was no meaningful difference in the baseline sociodemographic, clinical characteristics, and caries risk status data between the two groups (Table 1). No severe WSLs were observed in our patients. A low frequency of sugary and acidic food consumption between meals was observed in the patients with SSL and EL. In the SSL group, the median (range) score of sugar exposure between meals was 0.2 (0–2.8) times/day at T1. The median (range) score of acidic food consumption between meals was 0 (0–2.0) times/day at T1. Similarly, a low frequency of sugary and acidic food consumption was found in the EL group. At baseline, the patients were exposed to sugar (median, 0 times/day; range, 0–1.4 times/day). The median and range of acid exposure between meals were 0 (0–1.6) times/day in the EL group.

The questionnaire data concerning the patients’ medical history and dental health behaviors were collected (Table 2). None of the patients had systemic diseases or dry mouth-associated symptoms. Almost all of our patients (94.28%) performed tooth brushing at least twice a day. Tooth brushing was performed most frequently in the morning, after lunch, before bedtime, and at other times (>90%). Specific tooth brushing methods, i.e., horizontal, vertical, and circular scrubbing, were found in both groups (11.43%, SSL group and 20%, EL group), whereas the combined brushing method was the most common in both groups (88.57%, SSL group and 80%, EL group). Furthermore, 22.86% of the SSL group and 14.29% of the EL group spent less than 2 min brushing their teeth. In addition to brushing, 74.29% of the SSL group used dental floss, compared with 82.86% in the EL group. We found that 71.43% of patients with SSLs used interdental brushes, while this rate was 82.86% for the patients with ELs. Our results indicated that 57.14% of the SSL group and 74.29% of the EL group used mouthwash. Furthermore, 8.58% of the SSL group used other oral care products compared with 20% for the EL group. The frequency of professional fluoride application was 11.43% and 22.86% in the SSL and EL groups, respectively. 

### 3.2. Effects of Toothbrushing on Dental Biofilm Maturity

The ICC value demonstrated the high reliability of the PMS measurement (0.97). In the SSL group at T1, the % PMS of new dental biofilm (median 50%; Q1, 32.66%; Q3, 71.13%) was the highest compared with that of mature dental biofilm (median 26%; Q1, 20.42%; Q3, 47.93%) and cariogenic dental biofilm (median 8%; Q1, 4%; Q3, 22.92%) (*p* < 0.05) (Figure 3). After the intervention (T2), the % PMS of new dental biofilm in the SSL group significantly increased (median 79.16%; Q1, 62.96%; Q3, 96.15%) (*p* < 0.05). In contrast, the % PMS of mature dental biofilm (median 20%; Q1, 1.85%; Q3, 28.42%) and cariogenic dental biofilm (median 0%; Q1, 0%; Q3, 8.01%) in this group were significantly decreased (*p* < 0.05). Moreover, the Wilcoxon signed-rank test demonstrated a significant difference between the % PMS of immature, mature, and cariogenic dental biofilm in the SSL group at T2 (*p* < 0.05). 

The EL group demonstrated the highest % PMS of cariogenic dental biofilm (median 54.16%; Q1, 44.44%, Q3, 69.56%) at T1, while the median % PMS was 25.0% (Q1, 18.18%; Q3, 36.18%) and 13.63% (Q1, 7.55%; Q3, 26.96%) for mature and new dental biofilm, respectively (*p* < 0.05) (Figure 4). The EL group demonstrated a significant decrease in the % PMS of cariogenic dental biofilm (median 7.14%; Q1, 0; Q3, 15.78%), whereas a significant increase in that of immature dental biofilm (median 64%; Q1, 40.40%; Q3, 82.60%) was detected at T2 (*p* < 0.05). However, there was no significant difference in the % PMS of mature dental biofilm (median 29.62%; Q1, 13.04%; Q3, 46.21%) between T1 and T2 (*p* > 0.05). 

The change in the PMS (T1–T2) was compared between the SSL and EL groups (Figure 5). The result indicated that the % PMS change in immature dental biofilm in the SSL group was significantly greater than that in the EL group (*p* < 0.05). In contrast, the % PMS change in cariogenic dental biofilm in the SSL group was significantly less than that in the EL group (*p* < 0.05). However, there was no significant difference in the % PMS change in mature dental biofilm between the two groups (*p* > 0.05).

### 3.3. Benefits and Harms

The patients in this trial might benefit from the intervention by achieving better oral health. Before the interventions and all post-treatment visits, a comprehensive assessment of the teeth and oral soft tissues was conducted via visual examination of the oral cavity. Care, as usual, was provided following the standard orthodontic protocols. No adverse events were noted or reported during the study. No serious harm was observed.

## 4. Discussion

In this study, the confounders of dental caries were well-controlled between the two groups at baseline [35]. Toothbrushing is the most common oral hygiene practice for mechanical plaque control. It is a fundamental behavior for maintaining and promoting oral health [36]. The American Dental Association states that powered and manual toothbrushes are both effective at removing the dental biofilm that causes dental caries [37]. However, currently, there is no consensus on the recommendations for manual toothbrushing techniques between dentists, oral health therapists, and dental products [15]. Therefore, it is important to define toothbrushing techniques and to determine which techniques are more effective in removing dental biofilm to provide consistent and evidence-based oral hygiene education. Notably, the complete removal of dental biofilm can never be achieved, and after a single self-performed brushing, the amount of oral biofilm can only be reduced by 50–60% [38,39]. Fixed orthodontic appliances caused significant changes in the distribution of the dental biofilm. An increase in the amount of supragingival dental biofilm not only on the interproximal but also on the vestibular surface of the teeth was markedly observed in orthodontic patients [40]. In orthodontic patients, the number of locations out of reach of mechanical removal is higher, making orthodontic patients more prone to oral diseases than non-orthodontic patients [2]. In the present study, we found that the combined horizontal–Charters–modified Bass tooth brushing technique decreased the cariogenic dental biofilm in the SSL and EL groups to 8% and 25%, respectively. This implies that the toothbrushing technique effectively reversed the imbalance in dental biofilm in orthodontic patients [4].

Based on the ecological plaque hypothesis, caries preventive strategies should focus on reducing the cariogenic virulence factors that are responsible for dental biofilm dysbiosis [41]. However, previous studies on toothbrushing effectiveness typically monitored the quantity of dental biofilm rather than dental biofilm cariogenicity. Acid production at low pH is an important trait of cariogenic bacteria. Therefore, an assessment of acid production by the cariogenic bacteria in the biofilm can be an adjunctive indicator for caries risk assessment.

Several tests have been developed to detect the presence of acidogenic bacteria in dental biofilm. Colorimetric tests for chairside use provide a simple means for assessing the pH-lowering potential of dental biofilm without emphasizing any one particular organic acid [21,42]. Our results indicated that we could use a three-tone disclosing gel to monitor the effectiveness of toothbrushing in reducing dental biofilm cariogenicity. Currently, there is a lack of sufficient scientific evidence to support any recommendations on toothbrushing techniques for orthodontic patients [16]. A previous study compared the effectiveness of different toothbrushing methods in orthodontic patients [17]. They found a significant reduction in the Silness and Loe plaque index in orthodontic patients when using the scrubbing (46.1%), the modified Stillman (41.7%), and the Bass (32.5%) techniques for 9 months. In contrast, Bhatia et al. [18] reported that there was no significant difference in the Silness and Loe plaque index and *S. mutans* levels in dental biofilm in patients treated with fixed appliances after using the modified Bass toothbrushing technique for 12 months. However, they found a significant difference in the plaque index and *S. mutans* levels between the modified Bass and habitual technique groups. Our findings demonstrated the effectiveness of the horizontal technique combined with the Charter’s and modified Bass techniques based on the reduced dental biofilm cariogenicity in the EL and SSL groups.

Mature dental biofilm is closely related to periodontal diseases [43]. However, combined toothbrushing reduced the percentage of mature dental biofilm in the SSL but not the EL group. These differing results might be due to the differences in new retentive areas and the physical properties of the two ligatures. A rough surface provides a favorable environment for bacterial adhesion and biofilm maturation because a rough surface plays a protective role against shear force and increases the area available for biofilm formation [44]. SSL had pits, grooves, and scratches on the surface after being used in the oral cavity for four weeks [45] and two months [46]. In addition, a positive correlation was found between surface roughness and oral microbial adhesion on different orthodontic wires [45]. EL is considered an organic material, which would be more amenable to bacterial colonization than SSL, which is an inorganic material with an inert metal surface. The differences in surface topography and structural characteristics of EL and SSL may also enhance bacterial colonization [47].

A key strength of our study is that the CONSORT guideline was used to improve the quality of this trial [24]. Another strength is that the short-term nature of this trial was useful for controlling confounding variables, such as participant compliance. We also standardized our methodology by adhering to the combined horizontal–Charters–modified Bass toothbrushing technique. In addition, we assessed dietary habits as a possible confounder because dental biofilm formation is directly affected by diet habits and can vary depending on the types of foods consumed and oral hygiene practices [48,49].

Currently, for time-efficient orthodontics, elastic rings are used as the material of choice to ligate orthodontic arch wires to brackets rather than stainless steel ligatures. However, our results suggest that stainless steel wire ligation may be better than elastomeric ligation for dental biofilm management in orthodontic patients. A systematic review suggested that large, well-designed, randomized controlled trials are required to provide unequivocal recommendations for a particular ligation method that can effectively reduce dental biofilm formation in patients wearing fixed orthodontic appliances [5].

This study has some limitations. Blinding of the investigator and participants was not feasible because the difference in ligation materials was recognizable. To limit the potential for bias due to unblinded assessment, a clinical study suggested that the trial needed to modify the definition of the study outcome to be objective rather than subjective [50]. In the present study, dental biofilm maturity was objectively assessed because the different staining colors were distinctly discernable by the investigators and participants. In addition, the differences in the baseline PMS scores between the groups in this trial might affect the follow-up scores after the intervention. Therefore, the chance of bias was reduced by adjusting the statistical analysis for the baseline variables. The change scores were used to assess the effects of the assigned toothbrushing on the PMS in the SSL and EL groups.

## 5. Conclusions

Toothbrushing with a combined horizontal–Charters–modified Bass technique significantly decreased the percentage of cariogenic dental biofilm in orthodontic patients with SSL or EL. This combined toothbrushing method should be recommended for orthodontic patients who have a high amount of cariogenic dental biofilm. Our study demonstrates the effective combined toothbrushing practice for preventing carious lesion development in orthodontic patients.

## Figures and Tables

**Figure 1 healthcare-11-00731-f001:**
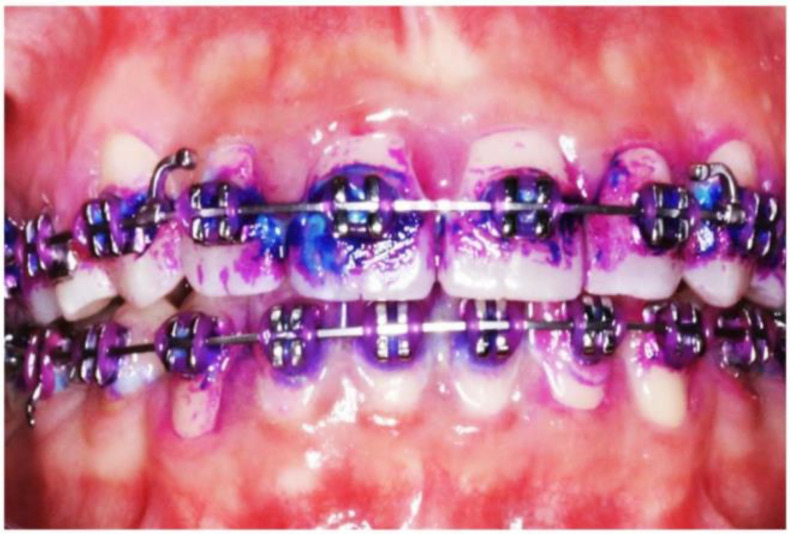
Common dental biofilm-related issues in an orthodontic patient. Cariogenic dental biofilm that is stained light blue is seen on 11 and 12. Mature dental biofilm is stained dark purple-blue, and immature dental biofilm is stained pink.

**Figure 2 healthcare-11-00731-f002:**
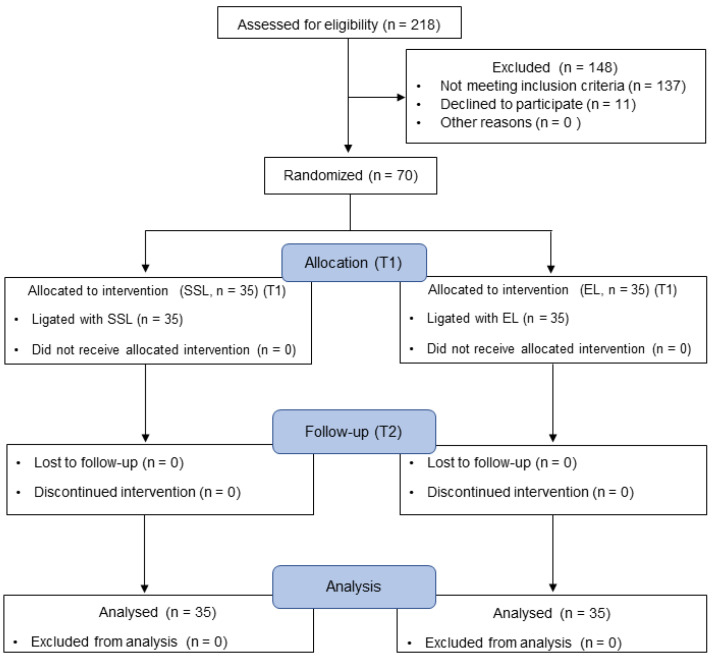
CONSORT flow diagram of the study participants.

**Figure 3 healthcare-11-00731-f003:**
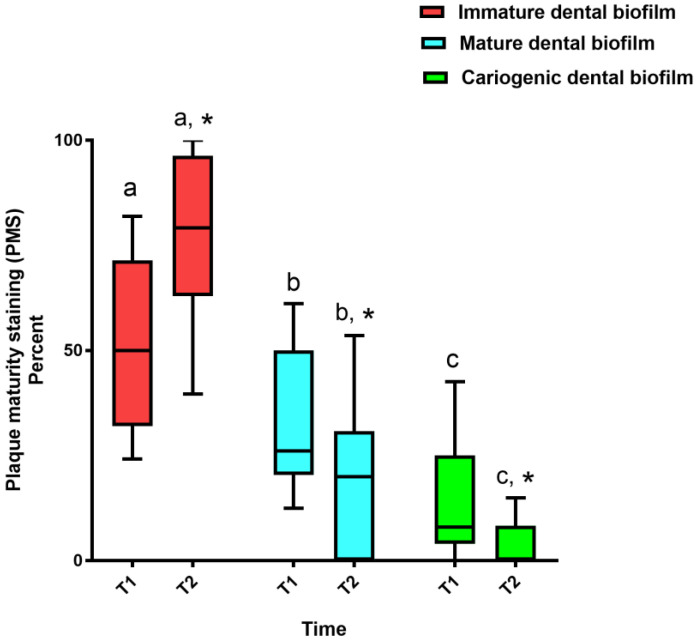
Frequency distribution of plaque maturity staining (PMS) in the SSL group. Dental biofilm staining was performed before SSL tying (T1) and at the 4-week follow-up (T2). The intervention was the combined horizontal–Charters–modified Bass tooth brushing technique. The boxes represent the interquartile range, and the horizontal bars within the boxes represent the median. The lower and upper ends of the vertical lines represent the 10th and 90th percentiles, respectively. The asterisks denote a significant difference in % PMS between T1 and T2 (Wilcoxon signed-rank test, *p* < 0.05). Different superscript letters indicate significant differences between the PMS at each time point (Kruskal–Wallis with Dunn–Bonferroni post hoc test, *p* < 0.05).

**Figure 4 healthcare-11-00731-f004:**
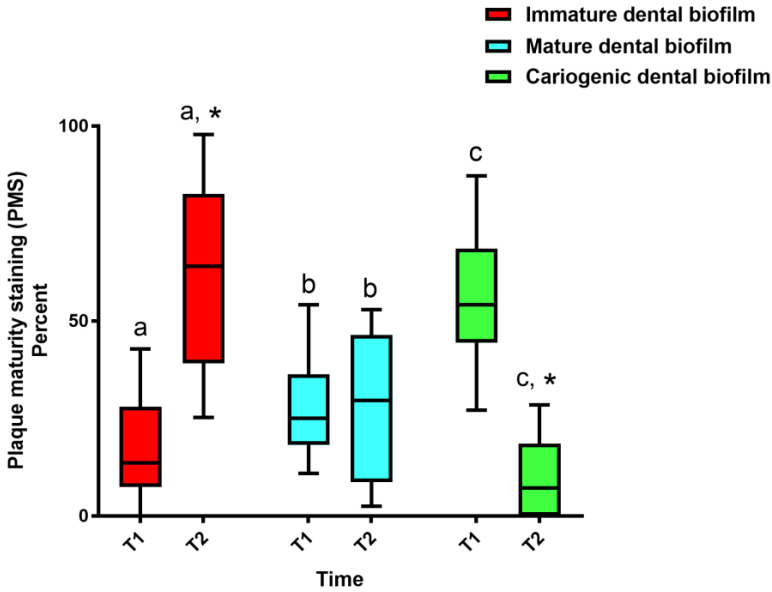
Frequency distribution of plaque maturity staining (PMS) in the EL group. Dental biofilm staining was performed before EL tying (T1) and at the 4-week follow-up (T2). The intervention was the combined horizontal–Charters–modified Bass tooth brushing technique. The boxes represent the interquartile range, and the horizontal bars within the boxes represent the median. The lower and upper ends of the vertical lines represent the 10th and 90th percentiles, respectively. The asterisks denote a significant difference in % PMS between T1 and T2 (Wilcoxon signed-rank test, *p* < 0.05). Different superscript letters indicate significant differences between the PMS at each time point (Kruskal–Wallis with Dunn–Bonferroni post hoc test, *p* < 0.05).

**Figure 5 healthcare-11-00731-f005:**
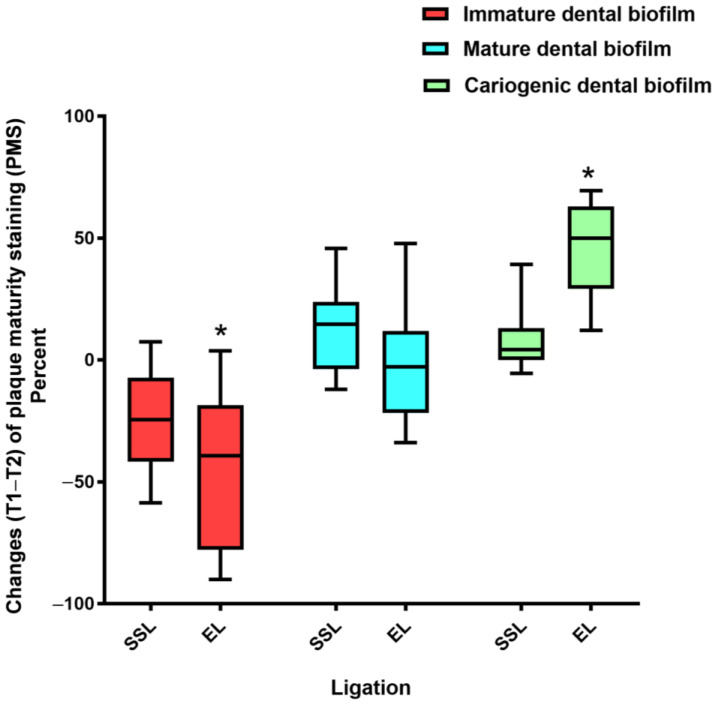
Comparison of plaque maturity staining (PMS) change before ligation (T1) and at the 4-week follow-up (T2) between the SSL and EL groups. Dental biofilm staining was performed at T1 and T2. The intervention was the combined horizontal–Charters–modified Bass tooth brushing technique. The boxes represent the interquartile range, and the horizontal bars within the boxes represent the median. The lower and upper ends of the vertical lines represent the 10th and 90th percentiles, respectively. The asterisks denote a significant difference in % PMS changes between the SSL and EL groups (Mann–Whitney U test, *p* < 0.05).

**Table 1 healthcare-11-00731-t001:** Distribution of the sociodemographic, WSL, and dietary habit data based on ligature type at baseline.

Variable	SSL	EL
Socio-demography		
Age (years)		
Median	28	28
Min–max	13–43	15–53
Sex		
Male, *n* (%)	5 (14.29)	8 (22.86)
Female, *n* (%)	30 (85.71)	27 (71.14)
Educational level, *n* (%)		
<Diploma	11 (31.43)	9 (25.71)
≥Diploma	24 (68.57)	26 (74.29)
White spot lesion (%)		
No [Median (min–max)]	100 (83.33–100)	96.42 (76.16–100)
Slight [Median (min–max)]	0 (0–16.16)	3.58 (0–16.16)
Severe [Median (min–max)]	0 (0–4.16)	0 (0–4.16)
Sugary intake between meals (time/day)		
Median	0.2	0.2
Min-max	0–2.8	0–1.4
Food acid intake between meals (time/day)		
Median	0	0
Min–max	0–2.0	0–1.6

**Table 2 healthcare-11-00731-t002:** Distribution of the patients’ medical history and dental health behaviors based on ligature type at baseline.

Variable	SSL	EL
Systemic disease, *n* (%)		
Yes	0 (0)	0 (0)
No	35 (100)	35 (100)
Medication-induced hyposalivation, *n* (%)		
Yes	0 (0)	0 (0)
No	35 (100)	35 (100)
Dry mouth, *n* (%)		
Yes	0 (0)	0 (0)
No	35 (100)	35 (100)
Difficulty swallowing, *n* (%)		
Yes	0 (0)	0 (0)
No	35 (100)	35 (100)
Feeling thirsty, *n* (%)		
Yes	0 (0)	0 (0)
No	35 (100)	35 (100)
Toothbrushing frequency, *n* (%)		
<2 times a day	2 (5.72)	2 (5.72)
≥2 times a day	33 (94.28)	33 (94.28)
Occasion of toothbrushing, *n* (%)		
In the morning	34 (97.14)	32 (91.43)
After lunch	34 (97.14)	32 (91.43)
Before bedtime	34 (97.14)	32 (91.43)
Other times	32 (91.43)	31 (88.57)
Toothbrushing technique, *n* (%)		
Horizontal, vertical, or circular	4 (11.43)	7 (20)
Combined	31 (88.57)	28 (80)
Toothbrushing time, *n* (%)		
<2 min	8 (22.86)	5 (14.29)
≥2 min	27 (77.14)	30 (85.71)
Use of dental floss, *n* (%)		
Yes	26 (74.29)	29 (82.86)
No	9 (25.71)	6 (17.14)
Use of interdental brush, *n* (%)		
Yes	25 (71.43)	33 (94.29)
No	10 (28.57)	2 (5.71)
Use of other oral healthcare products, *n* (%)		
Yes	3 (8.58)	7 (20)
No	32 (91.42)	28 (80)
Use of mouthwash, *n* (%)		
Yes	20 (57.14)	26 (74.29)
No	15 (42.86)	9 (25.71)
Professional fluoride application, *n* (%)		
Yes	4 (11.43)	8 (22.86)
No	31 (88.57)	27 (77.14)

## Data Availability

Not applicable.

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
