# Peer review of "Effectiveness of a Combined Toothbrushing Technique on Cariogenic Dental Biofilm in Relation to Stainless Steel and Elastomeric Ligatures in Orthodontic Patients: A Randomized Clinical Trial"

_healthcare, 2023, doi:10.3390/healthcare11050731_

Round 1

Reviewer 1 Report

This study is very important because it plays a role in oral hygiene in orthodontic treatment.

(1) First, compare your recommended Horizontal-Charters-modified Bass technique to conventional toothbrushing.

(2 )What is the basis for high cariogenic with 3 color identification dyes? It is thought that periodontal disease bacteria increase when plaque becomes mature?

(3)It seems necessary to describe the difference between biofilm and plaque in the text. Is there any difference in toothbrush between subjects in this experiment? I wonder if the time to brush my teeth is unified again?

(4)References 37 and 38 explain that it is never possible to completely remove dental biofilm, and that after a single self-brushing, the amount of oral biofilm can only be reduced by 50-60%. However, in this experiment, is this study compared under the same conditions such as time, pressure on the tooth surface, and toothbrush?

(5)The biggest drawback of this experiment is that the blinding of researchers and participants cannot be performed. How do you overcome this point?

Reviewer 2 Report

The manuscript presents the evaluation of the effect of a combined toothbrushing method on dental biofilm cariogenicity in patients with stainless steel (SSL) and elastomeric (EL) ligatures.

The topic is of practical significance.

The formatting is problematic as the subheadings are not left-aligned, and the paragraph indent is not consistent. Figure captions should be left-aligned. Appears the authors followed some publication protocols that contained certain section. The not applicable section like 2.6 should simply be eliminated.

Other than this, the manuscript is well-prepared.

In conclusion, a minor revision is needed before further consideration in JCS.

More details for consideration:

1. L423, authors stated “This discrepancy in our results might be due to the differences in new retentive areas and the physical properties of the two ligatures.” This needs to be further elucidated. Also especially that despite the EL group’s better dental health behaviors, e.g. higher brushing time, use of dental floss, interdental brush, other dental health products, mouthwash, fluoride application and the combined toothbrushing, etc. did not reduce the percentage of mature dental biofilm in this group.

Suggest adding a small section discussing surface area, surface characteristics such as roughness, hardness, bacterial adhesion, etc. Authors may cite relevant literature doi: 10.4012/dmj.2016-206; 10.1016/j.dental.2010.04.007.

2. This then begs the question, is SSL ligature the preferred practice?

3. Table 1. 2, some numbers are not all aligned.

Round 2

Reviewer 1 Report

The authors have responded appropriately to my comments. I think this paper is suitable for this journal.